# DOMAIN-SLOT RELATIONSHIP MODELING USING A PRE-TRAINED LANGUAGE ENCODER FOR MULTI-DOMAIN DIALOGUE STATE TRACKING

## ABSTRACT

Dialogue state tracking for multi-domain dialogues is challenging because the model should be able to track dialogue states across multiple domains and slots. Past studies had its limitations in that they did not factor in the relationship among different domain-slot pairs. Although recent approaches did support relationship modeling among the domain-slot pairs, they did not leverage a pre-trained language model, which has improved the performance of numerous natural language tasks, in the encoding process. Our approach fills the gap between these previous studies. We propose a model for multi-domain dialogue state tracking that effectively models the relationship among domain-slot pairs using a pre-trained language encoder. Inspired by the way the special $[CLS]$ token in BERT is used to aggregate the information of the whole sequence, we use multiple special tokens for each domain-slot pair that encodes information corresponding to its domain and slot. The special tokens are run together with the dialogue context through the pre-trained language encoder, which effectively models the relationship among different domain-slot pairs. Our experimental results show that our model achieves state-of-the-art performance on the MultiWOZ-2.1 and MultiWOZ-2.2 dataset.

## 1 INTRODUCTION

A task-oriented dialogue system is designed to help humans solve tasks by understanding their needs and providing relevant information accordingly. For example, such a system may assist its user with making a reservation at an appropriate restaurant by understanding the user's needs for having a nice dinner. It can also recommend an attraction site to a travelling user, accommodating the user's specific preferences. Dialogue State Tracking (DST) is a core component of these task-oriented dialogue systems, which aims to identify the state of the dialogue between the user and the system. DST represents the dialogue state with triplets of the following items: a domain, a slot, a value. A set of {*restaurant, price range, cheap*}, or of {*train, arrive-by, 7:00 pm*} are examples of such triplets. Fig. 1 illustrates an example case of the dialogue state during the course of the conversation between the user and the system. Since a dialogue continues for multiple turns of utterances, the DST model should successfully predict the dialogue state at each turn as the conversation proceeds. For multi-domain conversations, the DST model should be able to track dialogue states across different domains and slots.

Past research on multi-domain conversations used a placeholder in the model to represent domain-slot pairs. A domain-slot pair is inserted into the placeholder in each run, and the model runs repeatedly until it covers all types of the domain-slot pairs. (Wu et al., 2019; Zhang et al., 2019; Lee et al., 2019). A DST model generally uses an encoder to extract information from the dialogue context that is relevant to the dialogue state. A typical input for a multi-domain DST model comprises a sequence of the user's and the system's utterances up to the turn $t$, $X_t$, and the domain-slot information for domain $i$ and slot $j$, $D_iS_j$. In each run, the model feeds the input for a given domain-slot pair through the encoder.

$$f_{encoder}(X_t, D_iS_j) \text{ for } i = 1, \cdots, n, \quad j = 1, \cdots, m, \tag{1}$$

where $n$ and $m$ is the number of domains and slots, respectively. However, because each domain-slot pair is modeled independently, the relationship among the domain-slot pairs can not be learned. For example, if the user first asked for a hotel in a certain place and later asked for a restaurant near that hotel, sharing the information between {*hotel, area*} and {*restaurant, area*} would help the model recognize that the restaurant should be in the same area as the hotel.

Recent approaches address these issues by modeling the dialogue state of every domain-slot pair in a single run, given a dialogue context (Chen et al., 2020; Le et al., 2019). This approach can be represented as follows:

$$f_{encoder}(X_t, D_1 S_1, \cdots, D_n S_m). \qquad (2)$$

Because the encoder receives all of the domain-slot pairs, the model can factor in the relationship among the domain-slot pairs through the encoding process. For the encoder, these studies used models that are trained from scratch, without pre-training. However, since DST involves natural language text for the dialogue context, using a pre-trained language model can help improve the encoding process. Several studies used BERT (Devlin et al., 2019), a pre-trained bidirectional language model, for encoding the dialogue context (Zhang et al., 2019; Lee et al., 2019; Chao & Lane, 2019; Gao et al., 2019), but did not model the dependencies among different domain-slot pairs. Our approach fills the gap between these previous studies. In this work, we propose a model for multi-domain dialogue state tracking that effectively models the relationship among domain-slot pairs using a pre-trained language encoder. We modify the input structure of BERT, specifically the special token part of it, to adjust it for multi-domain DST.

The $[CLS]$ token of BERT (Devlin et al., 2019) is expected to encode the aggregate sequence representation as it runs through BERT, which is used for various downstream tasks such as sentence classification or question answering. This $[CLS]$ token can also be used as an aggregate representation for a given dialogue context. However, in a multi-domain dialogue, a single $[CLS]$ token has to store information for different domain-slot pairs at the same time. In this respect, we propose to use multiple special tokens, one for each domain-slot pair. Using a separate special token for each domain-slot pair is more effective in storing information for different domains and slots since each token can concentrate on its corresponding domain and slot. We consider two different ways to represent such tokens: *DS-merge* and *DS-split*. *DS-merge* employs a single token to represent a single domain-slot pair. For example, to represent a domain-slot pair of {*restaurant, area*}, we use a special token $DS_{(restaurant,area)}$. *DS-split*, on the other hand, employs tokens separately for the domain and slot and then merges them into one to represent a domain-slot pair. For {*restaurant, area*}, the domain token $D_{restaurant}$ and the slot token $S_{area}$. is computed separately and then merged. We use $\{DS\}_{merge}$ and $\{DS\}_{split}$ to represent the special tokens for *DS-merge* or *DS-split*, respectively. Unless it is absolutely necessary to specify whether the tokens are from *DS-merge* or *DS-split*, we'll refer to the DS-produced tokens as $\{DS\}$ tokens, without special distinction, in our descriptions forward. The $\{DS\}$ tokens, after being encoded by the pre-trained language encoder along with the dialogue context, is used to predict its corresponding domain-slot value for a given dialogue context.

| Turns | Utterances | Dialogue State |
|---|---|---|
| Turn 1 | System: 
 User: I am looking for a place to stay that has a cheap price range and it should be in a type of hotel | {hotel, price range, cheap}, {hotel, type, hotel} |
| Turn 2 | System: Okay, do you have a specific area you want to stay in? 
 User: No, I just need to make sure it's cheap. Oh, and I need parking | {hotel, price range, cheap}, {hotel, type, hotel}, {hotel, parking, yes} |
| Turn 3 | System: I found 1 cheap hotel for you that includes parking. Do you like me to book it? 
 User: Yes please. 6 people 3 nights starting on Tuesday | {hotel, price range, cheap}, {hotel, type, hotel}, {hotel, parking, yes}, {hotel, book day, Tuesday}, {hotel, book people, 6}, {hotel, book stay, 3} |

Figure 1: An example of a dialogue and its dialogue state.

## 2  RELATED WORKS

Recent work on dialogue state tracking can be largely divided into two groups according to how the slot-values are predicted: fixed-vocabulary and open-vocabulary. The fixed-vocabulary approach, also known as the picklisted-based approach, uses a classification module to predict the dialogue state for each slot from a pre-defined set of candidate values (Zhong et al., 2018; Nouri & Hosseini-Asl, 2018; Ramadan et al., 2018; Eric et al., 2019; Lee et al., 2019; Chen et al., 2020). The open-vocabulary approach generates the dialogue state for each domain-slot pair either by using a generative decoder to generate text (Wu et al., 2019; Hosseini-Asl et al., 2020) or by extracting text spans from the dialogue history (Gao et al., 2019; Goel et al., 2019; Heck et al., 2020). There is also an approach to use both picklist-based and span-based methods according to the slot type (Zhang et al., 2019).

For models that deal with multi-domain dialogue, how they deal with different domain-slot pairs is another way to divide them. The first approach encodes the dialogue context independent of the domain-slot pairs and uses separate modules for each domain-slot pair (Eric et al., 2019; Gao et al., 2019; Goel et al., 2019; Heck et al., 2020). The second approach encodes the dialogue context using the domain-slot pair information as the prefix and run the encoder multiple times (Nouri & Hosseini-Asl, 2018; Wu et al., 2019). Other approaches encode the dialogue context independently but merges it with domain-slot pair information later with a separate fusion module (Zhong et al., 2018; Ramadan et al., 2018; Lee et al., 2019). However, none of these models are able to model the relationship among different domain-slot pairs because there is no module that enables the interaction between them.

(Le et al., 2019) and (Chen et al., 2020) directly models the relationship among different domain-slot pairs. (Le et al., 2019) uses a Fertility decoder to learn potential dependencies across domain-slot pairs, but without using a pre-trained language model. Also, their model requires additional data such as system action and delexicalized system responses for its performance. (Chen et al., 2020) also explicitly models the relationship among different domain-slot pairs by using a Graph Attention Network (GAT) (Veličković et al., 2018). Schema graphs, which is the relation graph between domains and slots, are utilized for connecting edges in the GAT. Our work is different from these works in that we leverage the power of a pre-trained language encoder for directly modeling the dependencies among different domain-slot pairs.

(Hosseini-Asl et al., 2020) takes a different approach from the others by using multi-task learning that encompasses DST as well as action and response generation with a generative language model GPT-2 (Radford et al., 2019). However, since our work is focused on DST, we consider the model that is trained on DST only. In the decoding process, dialogue states for different domain-slot pairs are sequentially generated.

## 3  PROPOSED METHOD

Our model is composed of three parts. The first is the domain-slot-context (DSC) encoder, which encodes the dialogue context along with the special tokens representing domain-slot pairs. Next is slot-gate classifier, which is a preliminary classifier that predicts whether each domain-slot pair is relevant to the dialogue context. The adopted the concept of the slot-gate classifier from (Wu et al., 2019) and made adjustments to apply to our model. The last is the slot value classifier for predicting the value for each domain-slot pair among the candidate values.

In the following descriptions, we assume a dialogue context with a total of $T$ turns. The task is to predict the dialogue state, which are {*domain, slot, value*} triplets for all domain-slot pairs, for every turn $t = 1, \cdots, T$, using the dialogue context until each turn. Section 3 show the overview of our proposed model.

### 3.1  DOMAIN-SLOT-CONTEXT ENCODER

The main structure of our model is the DSC encoder, which uses a pre-trained language to encode the dialogue context along with $\{DS\}$ tokens. For the pre-trained language encoder, we used ALBERT (Lan et al., 2019) due to its strong performance on numerous natural language understanding tasks while having fewer parameters compared to other BERT-style encoders. $\{DS\}$ tokens work like

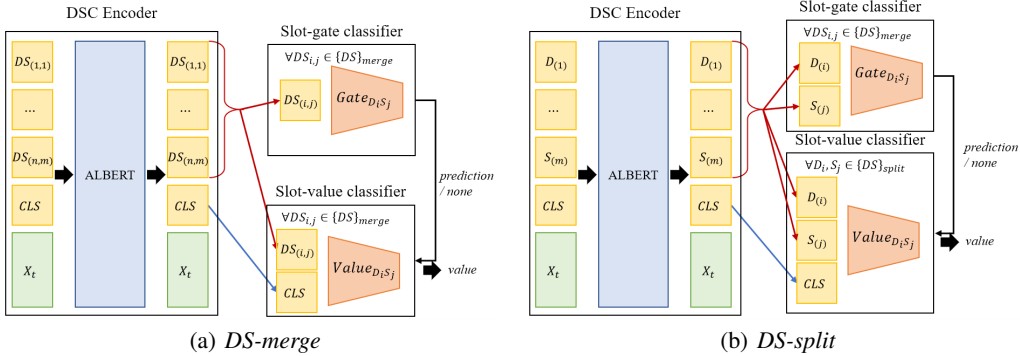

Figure 2: Model overview.

the $[CLS]$ token for BERT, encoding information corresponding to its domain-slot pair (*DS-merge*) or domain and slot (*DS-split*). The set of special tokens for each layout are shown in Eq. (3) and Eq. (4), respectively. In *DS-merge*, we used special tokens for each individual domain-slot pair. If there are many domain-slot pairs, using this layout can increase the number of special tokens as each domain-slot pair requires a separate special token. In *DS-split*, we used separate tokens for the domain and slot. To represent a domain-slot pair, we merged the corresponding tokens from each domain and slot by concatenating them. This promotes modeling compositionality, since the same slot token can be used for different domains. These $\{DS\}$ tokens and the dialogue context are processed through the DSC encoder, which results in each token in $\{DS\}$ being encoded with contextualized representations according to its domain and slot.

$$\{DS\}_{merge} = \{DS_{(domain_{(1)}, slot_{(1)})}, \cdots, DS_{(domain_{(n)}, slot_{(m)})}\} \tag{3}$$

$$\{DS\}_{split} = \{D_{domain_{(1)}}, \cdots, D_{domain_{(n)}}, S_{slot_{(1)}}, \cdots, S_{slot_{(m)}}\} \tag{4}$$

Fig. 3 shows the input representation of the DSC encoder. The sequence begins with $\{DS\}$ tokens. The special token $[CLS]$ follows, which encodes the overall information of the dialogue context. For the dialogue context, we added a special token $[SEP_u]$ to separate each user or system utterance, which is added at the end of each utterance from the user or system. The input ends with a special token $[SEP]$ as the end-of-sequence token.

4 types of embeddings are summed up to represent each token embedding. We used the pre-trained word embedding of ALBERT, except for the $\{DS\}$ tokens, which are randomly initialized. We introduced the token type embedding to differentiate the $\{DS\}$ tokens, user utterances tokens, and system utterances tokens. For *DS-merge*, we used a single token type embedding to represent a domain-slot pair, whereas for *DS-split*, we used two token type embeddings, one for the domain and the other for the slot. We did not apply this embedding for the $[CLS]$ token. Position embeddings are also employed from ALBERT, but the index of the positional embedding starts from the $[CLS]$ token. We did not use the positional embedding for the $\{DS\}$ tokens as the order within those tokens is meaningless. Lastly, the segment embedding from ALBERT was used to represent the whole sequence as a single segment, which is the default segment embedding of ALBERT.

DSC encoder encodes contextualized embeddings for every input token. However, for the slot-gate classifier and slot-value classifier, we only use the special token outputs of the DSC encoder ($[CLS]$ token and $\{DS\}$ tokens). This is formally defined as follows for *DS-merge* and *DS-split*, respectively, for turn $t$:

$$\widehat{DS_{(1,1)}}, \cdots, \widehat{DS_{(n,m)}}, \widehat{CLS} = DSCencoder([\{DS\}_{merge}, CLS, X^t, SEP]), \tag{5}$$

$$\widehat{D_1}, \cdots, \widehat{D_n}, \widehat{S_1} \cdots, \widehat{S_m}, \widehat{CLS} = DSCencoder([\{DS\}_{split}, CLS, X^t, SEP]), \tag{6}$$

where $X^t$ represents the dialogue context of $(S^1, SEP_u U^1, SEP_u, \cdots, S^t, SEP_u, U^t, SEP_u)$. $U^t$ and $S^t$ represents the utterance for the $t^{th}$ turn for the user and system respectively. The $\{DS\}$

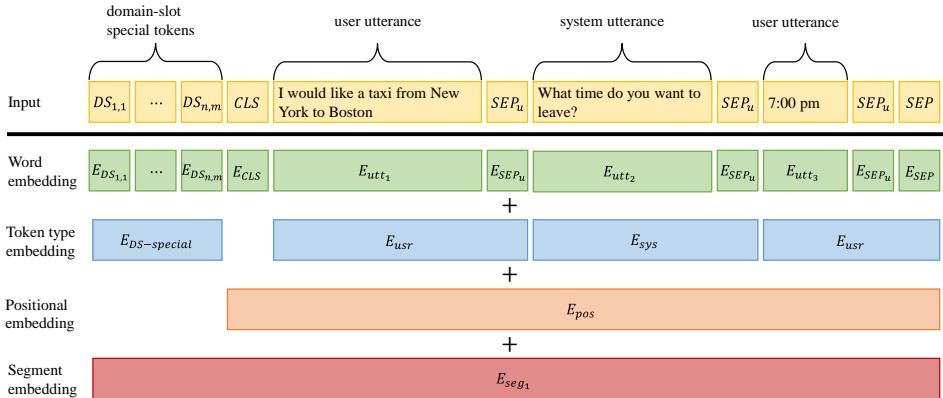

Figure 3: Input representation for the DSC encoder. The example here shows a dialogue context for 2 turns of $(S^1, U^1, S^2, U^2)$. $S^1$ is omitted because the sequence starts with the user utterance and $S^1$ is just a placeholder: a blank sentence. In this figure, the special token layout for $\{DS\}$ tokens is represented in *DS-merge*. $E_{utt_n}$ represents the sequence of word embeddings for each utterance after tokenization.

tokens and $[CLS]$ token with the hat notation $\hat{}$ represents the encoded output of the DSC encoder for those special tokens. They are vectors of $\mathbb{R}^d$, where $d$ is the hidden dimension of ALBERT.

## 3.2 SLOT-GATE CLASSIFIER

For the slot-gate classifier, we use the DSC encoder output of the $\{DS\}$ tokens for each domain-slot pair to predict whether it is relevant to the dialogue or not. In previous methods, gating used categories of $\{prediction, dontcare, none\}$, where $prediction$ means a slot value is not $dontcare$ or $none$ and $dontcare$ means that the predicted slot value is $dontcare$ and $none$ means that the domain-slot is non-relevant. The label for slot-gates are made from the slot-values. However, the performance for the $dontcare$ category was far inferior to the other two categories, so we dismissed the $dontcare$ category and only used $\{prediction, none\}$. In our preliminary models with ALBERT *large-v2*, the prediction and recall for $dontcare$ was $48.87\%$ and $17.21\%$, respectively. The precision and recall for $none$ showed $98.91\%, 99.45\%$ and $prediction$ $96.16\%, 94.93\%$, respectively. In this setting, the $dontcare$ category is included in $prediction$. For *DS-merge*, the slot-gate classifier predicts the value using the domain-slot pair special token. For the domain-slot pair of domain $i$ and slot $j$, the slot-gate classifier output for *DS-merge* is

$$Gate_{D_i S_j} = sigmoid\big(W_{G_{DS_{(i,j)}}} \widehat{DS_{(i,j)}}\big), \tag{7}$$

where $W_{G_{D_i S_j}} \in \mathbb{R}^{1 \times d}$. For *DS-split*, the slot-gate classifier uses the concatenated output of the corresponding domain and slot token. Similarly, for the same domain-slot pair, the slot-gate classifier output for *DS-split* is

$$Gate_{D_i S_j} = sigmoid\big(W_{G_{(D_i, S_j)}} \big[\widehat{D_i} | \widehat{S_j}\big]\big), \tag{8}$$

where | represents concatenation of vectors and $W_{G_{(D_i, S_j)}} \in \mathbb{R}^{1 \times 2d}$. The loss objective for the gate classification is as follows.set

$$\mathcal{L}_{gate} = \sum_{(i,j) \in DS} BinaryCrossEntropy\big(y^{gate}_{D_i S_j}, Gate_{D_i S_j}\big), \tag{9}$$

where $DS$ refers to the set of all domain-slot pairs and $y^{gate}_{D_i S_j}$ is the binary slot-gate label for domain $i$ and slot $j$. If the domain-slot is predicted to $none$, the corresponding output of the slot-value classifier is changed into $none$ regardless of the prediction of the slot-value classifier.

### 3.3 SLOT-VALUE CLASSIFIER

We employ the fixed-vocabulary based classification method for predicting slot values. As in (Zhang et al., 2019), the candidate-value list for each domain-slot pair was constructed by using the values from the training dataset, rather than using the incomplete ontology from the dataset. The $[CLS]$ token is concatenated with each token from $\{DS\}$, and used as the input to the slot-value classifier for each domain-slot pair. The slot-value classifier output of domain $i$ and slot $j$ for *DS-merge* is as follows:

$$Value_{D_i S_j} = softmax\big(W_{V_{DS_{(i,j)}}} \big[\widehat{DS_{(i,j)}} | \widehat{CLS}\big]\big), \tag{10}$$

where $W_{V_{DS_{(i,j)}}} \in \mathbb{R}^{n_{D_i S_j} \times 2d}$ and $n_{D_i S_j}$ is the number of candidate values for domain $i$ and slot $j$. Similarly, for *DS-split*, the slot-value classifier output is

$$Value_{D_i S_j} = softmax\big(W_{V_{(D_i, S_j)}} \big[\widehat{D_i} | \widehat{S_j} | \widehat{CLS}\big]\big), \tag{11}$$

where $W_{V_{(D_i, S_j)}} \in \mathbb{R}^{n_{D_i S_j} \times 3d}$. The loss objective for the slot-value classification is as follows:

$$\mathcal{L}_{value} = \sum_{(i,j) \in DS} CrossEntropy(y_{D_i S_j}^{value}, Value_{D_i S_j}), \tag{12}$$

where $y_{D_i S_j}^{value}$ is the label for domain $i$ and slot $j$.

### 3.4 TOTAL OBJECTIVE FUNCTION

The DSC encoder, slot-gate classifier and slot-value classifier is jointly trained under the total objective function below.

$$\mathcal{L}_{total} = \mathcal{L}_{gate} + \mathcal{L}_{value} \tag{13}$$

## 4 EXPERIMENT SETUP AND RESULTS

We evaluate our model using the joint goal accuracy, which considers a model prediction to be correct when the prediction jointly matches the ground truth values for all domain-slot pairs, given a dialogue context.

### 4.1 DATASET

We use the MultiWOZ-2.1 (Eric et al., 2019) and MultiWOZ-2.2 dataset (Zang et al., 2020), both of which fixed noisy annotations and dialogue utterances of the MultiWOZ 2.0 dataset (Budzianowski et al., 2018). The dataset contains 7 domains and over 10,000 dialogues. We follow the previous studies and use 5 domains (train, restaurant, hotel, taxi, attraction) with 30 domain-slot pairs. The other two domains (police, hospital) have little data and do not appear in the test dataset. For MultiWOZ-2.1, we follow the pre-processing explained in (Wu et al., 2019). For MultiWOZ-2.2, we use the raw data as given without any pre-processing.

### 4.2 SETUP

For the pre-trained language encoder, we used ALBERT(Lan et al., 2019) from HuggingFace (Wolf et al., 2019) in Pytorch (Paszke et al., 2019). We used the *xxlarge-v2* version of ALBERT for the main experiment and compare other versions (*base-v2*, *large-v2*) in the analysis section. We also compared RoBERTa (Liu et al., 2019) to generalizability of our model. The optimizer was AdamW (Loshchilov & Hutter, 2018) with a learning rate of $1e^{-5}$ for *ALBERT-xlarge-v2*, *ALBERT-xxlarge-v2* and *RoBERTa-large* and $5e^{-5}$ for *ALBERT-base-v2*, *ALBERT-large-v2* and *RoBERTa-base*. We applied linear warm-up followed by linear decay for the learning rate. We trained all models with the effective batch size of 32, using gradient accumulation for bigger ALBERT models. Models were

Table 1: Results for the test dataset of the Multi-WOZ 2.1 and MultiWOZ 2.2 dataset. *: extra supervision information is from (Hosseini-Asl et al., 2020).

| Model | Extra Supervision | MultiWOZ-2.1 | MultiWOZ-2.2 |
|---|---|---|---|
| SGD-baseline (Rastogi et al., 2020) | - | 43.4 | 42.0 |
| TRADE (Wu et al., 2019) | - | 46.0 | 45.4 |
| NADST (Le et al., 2019) | sys. action, delex. sys. response | 49.04 | |
| DSTQA (Zhou & Small, 2019) | knowledge graph* | 51.17 | |
| DS-DST (Zhang et al., 2019) | - | 51.20 | 51.7 |
| DS-Picklist (Zhang et al., 2019) | - | 53.30 | |
| SST (Chen et al., 2020) | - | 55.23 | |
| TripPy (Heck et al., 2020) | action decision* | 55.3 | |
| SimpleTOD (Hosseini-Asl et al., 2020) | - | 55.76 | |
| CHAN (Shan et al., 2020) | - | 58.55 | |
| ConvBERT-DG + Multi (Mehri et al., 2020) | DialoGLUE (Mehri et al., 2020) data | 58.7 | |
| Our work *(DS-merge, ALBERT-xxlarge-v2)* | - | **69.23** | **77.28** |
| Our work *(DS-split, ALBERT-xxlarge-v2)* | - | **67.31** | **82.23** |

selected based on their joint goal accuracy on the validation data split. Only the training data was used to build the labels for each domain-slot pair. We used two NVIDIA V100 for our training. The original ALBERT was pre-trained with a sequence length of up to 512 tokens. However, dialogues that are longer than 512 tokens exists in the data. Usually, the standard procedure for this situation is to truncate the sequence up to 512 tokens and discard the remaining tokens. However, to cover dialogues longer than 512 tokens that are in the dataset, we resized the positional embedding to cover a maximum length of the dialogue. We preserved the original pre-trained position embedding for positions indices up to 512 and randomly initialized the remaining position indices. This method showed better results than limiting the maximum sequence length to 512. We plan to release our code on Github.

## 4.3 RESULTS

Table 1 shows the joint goal accuracy of our model compared to previous methods. Both of our models show better performance among models without any additional supervision other than the dialogue context and domain-slot pair labels. Especially, the *DS-split, ALBERT-xxlarge-v2* version of our proposed model achieves state-of-the-art result on the MultiWOZ-2.1 and MultiWOZ-2.2 dataset, without any form of extra supervision. However, in smaller models, The model with *DS-split* shows better results than the model with *DS-merge*. This shows that in models with enough capacity, the slot-sharing of *DS-split* was more effective. However, this was not the case for smaller ALBERT models, which is explained in Section 4.4.2. This is important in that scalability is much better for *DS-split* than *DS-merge*, as many slots can be shared across different domains, reducing the number of special tokens to be used. We show the individual domain-slot accuracy in Appendix A.2, Table 4.

## 4.4 ANALYSIS

In this section, we show that relationship modeling among different domain-slot pairs is indeed the key factor of our proposed model by running ablation studies. Also, we compare the effect of the size and type of the pre-trained language encoder in terms of performance.

### 4.4.1 RELATIONSHIP MODELING AMONG DIFFERENT DOMAIN-SLOT PAIRS

First, we did not use any $\{DS\}$ tokens and only used the $CLS$ token. Because there are no dedicated special tokens for each domain-slot pair, the performance is very poor as shown in 'None' row in Table 2. This shows that our approach to introduce $\{DS\}$ is effective.

Next, to evaluate the effect of relationship modeling among different domain-slot pairs, we blocked the attention among different $\{DS\}$ tokens during the encoding process, which restricts direct interaction among $\{DS\}$ tokens. Table 2 shows that without the relationship modeling, our model performance deteriorates by a substantial amount. This validates our idea that relationship modeling is the crucial factor for our approach.

Table 2: Results for ablation of domain-slot relationship modeling on the test dataset of MultiWOZ 2.1.

| Pretrained Language Encoder | $\{DS\}$ token layout | Joint Goal Accuracy |
|---|---|---|
| *ALBERT-large-v2* | None | 45.49 |
| | *DS-merge* | 55.48 |
| |   w/o relationship modeling | 53.38 |
| | *DS-split* | 55.06 |
| |   w/o relationship modeling | 52.84 |
| *ALBERT-xxlarge-v2* | None | 50.96 |
| | *DS-merge* | 67.31 |
| |   w/o relationship modeling | 63.46 |
| | *DS-split* | 69.23 |
| |   w/o relationship modeling | 61.54 |

Table 3: Results for different ALBERT configurations on the evaluation test dataset of MultiWOZ 2.1.

| $\{DS\}$ token layout | Pretrained Language Encoder | Joint Goal Accuracy |
|---|---|---|
| *DS-merge* | *ALBERT-base-v2* | 55.01 |
| | *ALBERT-large-v2* | 55.48 |
| | *ALBERT-xlarge-v2* | 57.02 |
| | *ALBERT-xxlarge-v2* | 67.31 |
| | *RoBERTa-base* | 53.85 |
| | *RoBERTa-large* | 55.77 |
| *DS-split* | *ALBERT-base-v2* | 53.54 |
| | *ALBERT-large-v2* | 55.06 |
| | *ALBERT-xlarge-v2* | 55.51 |
| | *ALBERT-xxlarge-v2* | 69.23 |
| | *RoBERTa-base* | 57.69 |
| | *RoBERTa-large* | 58.65 |

In the Appendix A.1, we show some examples of wrong predictions that models without direct relationship modeling has made.

### 4.4.2 SIZE AND TYPE OF THE PRE-TRAINED LANGUAGE ENCODER

We compared ALBERT and RoBERTa (Liu et al., 2019) and various model sizes within those pre-trained language encoders. Table 3 shows the result for different versions of the pre-trained language encoders. For ALBERT, a bigger language model shows better results as is shown in various downstream tasks that ALBERT was evaluated on (Lan et al., 2019). Except for *ALBERT-xx-large*, all other configurations show that *DS-merge* shows better performance than *DS-split*. Based on the drastic increase in performance with *xx-large*, we presume that the high model complexity of *ALBERT-xx-large* enabled $\{DS\}_{split}$ tokens to effectively encode information and make slot-sharing to work. In smaller models, this slot-sharing might not have been as effective due to their smaller encoding capacity. Also, concatenation, which was used for merging domain and slot embeddings in *DS-split*, might not have been enough for fully representing the information for the domain-slot pair in smaller models. RoBERTa also shows similar results with bigger models showing stronger performance.

### 4.4.3 LEARNING CURVE

Fig. 4 shows the learning curve of the *ALBERT-xxlarge-v2* on the MultiWOZ-2.2 dataset. The joint goal accuracy steadily increases after the slot-value loss plateaus.

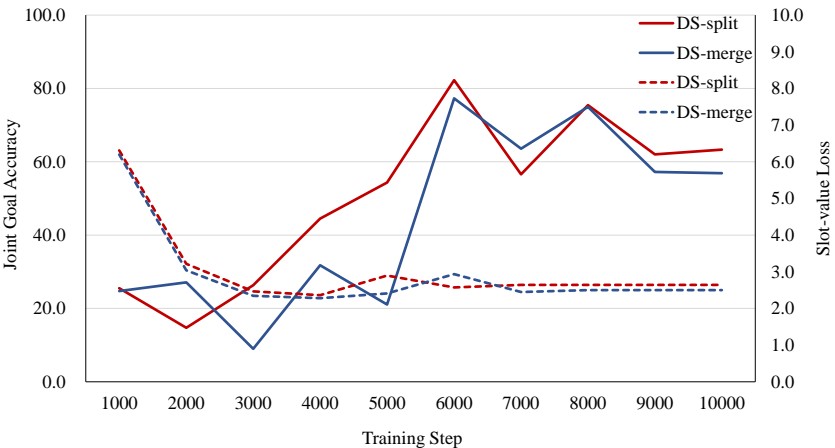

Figure 4: Joint goal accuracy and slot-value loss versus training steps on the evaluation dataset of MultiWOZ-2.2. *ALBERT-xxlarge-v2* was used for the DSC-encoder. Straight lines indicate the joint goal accuracy and dashed lines indicate the slot-value loss. The x-axis indicates the training steps.

## 5 CONCLUSION

In this paper, we propose a model for multi-domain dialogue state tracking that effectively models the relationship among domain-slot pairs using a pre-trained language encoder. We introduced two methods to represent special tokens for each domain-slot pair: *DS-merge* and *DS-split*. These tokens work like the $[CLS]$ token for BERT, encoding information corresponding to its domain-slot pair (*DS-merge*) or domain and slot (*DS-split*). These special tokens are run together with the dialogue context through the pre-trained language encoder, which enables modeling the relationship among different domain-slot pairs. Experimental results show that our model achieves state-of-the-art performance on the MultiWOZ-2.1 and MultiWOZ-2.2 dataset. The ablation experiments show that the relationship modeling among different domain-slot pairs is the key element of our model. Also, we showed that larger pre-trained language encoders improves performance. We hope to advance our research by finding ways to effectively apply our model towards the open-vocabulary approach, which will enable better generalization for candidate values that are outside of the training data.

### ACKNOWLEDGMENTS

This research was supported and funded by the Korean National Police Agency. [Pol-Bot Development for Conversational Police Knowledge Services / PR09-01-000-20]

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

## A   APPENDIX

### A.1   RELATIONSHIP MODELING EXAMPLES

#### A.1.1   EXAMPLE 1

Fig. 5 shows an example of a wrong prediction that the model without domain-slot relationship modeling makes. The value for {*taxi, departure*} is not explicitly mentioned in the dialogue context. However, Our full model correctly predicts the value for {*taxi, departure*}, which can be inferred from the dialogue context and {*hotel, name*}. However, the model without relationship modeling fails to predict the correct value for {*taxi, departure*}.

**User:** i am staying in cambridge soon and would like to stay at a and b guest house.

**System:** sure, how many days and how many people?

**User:** we are staying 6 people for 4 nights starting from tuesday. i need the reference number

**System:** your booking is successful! your reference number is iigra0mi. do you need anything else?

**User:** yeas, what to recommend if i want to see good architecture in the west part of town?

**System:** unfortunately there is no good architecture on the west end but i can look in other parts of town if you want

**User:** what about a museum?

**System:** what part of town there are none in the west.

**User:** there are no museums in the west at all?

**System:** sorry about that, there are actually 7 in that area.

**User:** great, can i get the postcode, entrance fee and address of 1 of them?

**System:** cafe jello gallery has a free entrance fee. the address is cafe jello gallery, 13 magdalene street and the post code is cb30af. can i help you with anything else?

**User:** yes please. i need a taxi to commute.

**System:** when would you like to leave and arrive?

**User:** i would like to get to the gallery by 13:45, please.

**System:** sure, lookout for a blue volvo the contact number is 07941424083. can i help with anything else?

**User:** that is all for now. thank you so much

| Domain-slot | Proposed method | w/o relationship modeling |
|---|---|---|
| hotel-name | a and b guest house | a and b guest house |
| hotel-book day | tuesday | tuesday |
| hotel-book day | 4 | 4 |
| attraction-area | Tuesday | Tuesday |
| attraction-type | west | west |
| taxi-destination | museum | museum |
| taxi-departure | cafe jello gallery | **none (WRONG)** |
| taxi-arriveby | 13:45 | 13:45 |

Figure 5: An example of a dialogue and its dialogue state.

### A.1.2   EXAMPLE 2

Fig. 6 also shows an example of a wrong prediction that the model without domain-slot relationship modeling makes. The value for {*train, day*} is not explicitly mentioned in the dialogue context. In a similar manner from the example above, it can be referred from the {*restaurant, book day*}.

**User:** i would like to find a particular restaurant in cambridge. the name of the restaurant is restaurant 2 two. could you give me the location?

**System:** restaurant 2 two is nice french restaurant located at 22 chesterton road chesterton. would like me to book you a table?

**User:** that would be great. i need it for 8 on friday.

**System:**  do you have a time preference?

**User:** yes at 11:15 if that is not available i can do 10:15

**System:** the booking for 10:15 was successful they will reserve the table for 15 minutes. the reference number is 6b5z7vj5.

**User:** thanks. can you help me find a train, too? i want to leave cambridge some time after 12:15.

| Domain-slot | Proposed method | w/o relationship modeling |
|---|---|---|
| train-day | friday | **none (WRONG)** |
| train-departure | cambridge | Cambridge |
| train-leaveat | 12:15 | 12:15 |
| restaurant-name | restaurant 2 two | restaurant 2 two |
| restaurant-book time | 10:15 | 10:15 |
| restaurant-book day | friday | friday |
| restaurant-book people | 8 | 8 |

Figure 6: An example of a dialogue and its dialogue state.

## A.2 INDIVIDUAL SLOT ACCURACY

Table 4 shows the individual domain-slot accuracy for the *ALBERT-xxlarge-v2* model on the MultiWOZ-2.2 dataset.

Table 4: Individual domain-slot accuracy for the *ALBERT-xxlarge-v2* model on the MultiWOZ-2.2 dataset.

| Domain-slot | DS-merge | DS-split |
|---|---|---|
| attraction-area | 94.56 | 96.81 |
| attraction-name | 94.43 | 96.62 |
| attraction-type | 94.4 | 96.92 |
| hotel-area | 95.41 | 97.75 |
| hotel-book day | 96.05 | 98.5 |
| hotel-book people | 96.32 | 98.52 |
| hotel-book stay | 95.97 | 98.5 |
| hotel-internet | 95.6 | 97.76 |
| hotel-name | 94.84 | 97.21 |
| hotel-parking | 95.7 | 98.1 |
| hotel-price range | 95.36 | 97.72 |
| hotel-stars | 94.96 | 97.34 |
| hotel-type | 95.47 | 97.35 |
| restaurant-area | 94.81 | 97.21 |
| restaurant-book day | 97.72 | 100 |
| restaurant-book people | 97.61 | 100 |
| restaurant-book time | 97.67 | 100 |
| restaurant-food | 95.1 | 97.21 |
| restaurant-name | 94.12 | 96.34 |
| restaurant-price range | 94.85 | 97.24 |
| taxi-arrive by | 97.19 | 99.42 |
| taxi-departure | 96.29 | 98.57 |
| taxi-destination | 96.63 | 98.73 |
| taxi-leave at | 97.05 | 99.28 |
| train-arrive by | 97.54 | 100 |
| train-book people | 95.49 | 97.78 |
| train-day | 93.03 | 95.14 |
| train-departure | 92.6 | 95.01 |
| train-destination | 92.73 | 94.84 |
| train-leave at | 94.87 | 97.08 |

