# OpenReview forum: "Domain-slot Relationship Modeling using  a Pre-trained Language Encoder for Multi-Domain Dialogue State Tracking"
_ICLR.cc/2021/Conference — Reject_

### Official Review · AnonReviewer2 · 2020-10-23

**Rating:** 4
**Confidence:** 5

**Review:**

[Summary]
In this paper, the authors proposed a multidomain state-tracking model that leverages the relationship among different domain-slot pairs. This is done by leveraging the full-attention step over the [CLS] special token and by providing all the domain-slot pairs as a special token to a pre-trained language model (Figure 2 is very clear). To predict the value of the slot $D_{i,j}$, the author concatenates the representation of the [CLS] token, share among all the domain-slots, and the $D_{i,j}$, provided as input, and use a gating mechanism, by only using $D_{i,j}$ representation, to decide whether require as value (i.e., prediction) or not (e.g. None). \

The authors experimented using ALBERT (Lan et al., 2019) as a pre-trained language model, on the well-known benchmark MultiWoZ 2.0  (Budzianowski et al., 2018)  2.1 (Eric et al., 2019). The authors studied different format to represent $D_{i,j}$ DS-merge (i.e., one token per domain-slot) and DS-split (i.e., one token per slot and one per domain, thus more scalable). The reported performance is state-of-the-art at the time of submission.

[Pros]
- The paper reads well and it is easy to follow for people working on Task-Oriented dialogue.
- The proposed method is simple and effective, and it would be easy to reproduce.

[Cons]
- The idea of using domain-pairs as input to a large pre-trained model is not novel (Wu et al., 2019; Zhang et al., 2019; Lee et al., 2019), as also pointed out by the authors, but the authors do not explicitly clarify this in the methodology section, leading the reader to believe that the domain-pairs is their own contribution. Same for the slot-gate (Wu et al., 2019)
- The authors claim to learn relations between slots, but the analysis section is very thin and it just shows an ablation by masking the attention between the slot. Two points: why not just removing the [CLS] token instead of removing the attention, and why just using on ALBERTA large. For instance, the authors said "For this experiment, we used the ALBERT configuration of large-v2,
for faster experimentation" which is contradictory since large-v2 is the slowest to run I guess. Can the authors show this ablation for all the model size?
- Although, MWoZ is the current benchmark for DST in ToDs, there are also other datasets for this task that can be considered (e.g., Schema Guided Dialogue (SGD) (Rastogi et.al. 2019))


[Reason to Reject]
The main contribution of this paper is very thin, adding the [CLS] token as input, and the main technical contribution is not well explored (missing an in-depth ablation).

[Reason to Accept]
State-of-the-art performance at the submission time. To be noted, (Mehri et.al. 2020) reported better performance in MWoZ and other datasets, but this paper was released after the ICLR submission deadline.

[Question]
-  Can the authors show the ablation for all the model size?

[Suggestion]
- Figure 4 is very hard to read. I suggest to better format the dialogue.

---

> ### Author Response · Authors · 2020-11-13
> **Author Response to AnonReviewer2**
>
> Thank you very much for your review of our paper. We initially updated our paper with an updated result using *xxlarge-v2* of ALBERT, which showed the state-of-the-art result. We are running additional experiments to respond to the reviewers comment, as soon as the experiments are done we will update the manuscript right away.
>
> Now, we will try to explain each point that you have mentioned.
>
> 1. The idea of using domain-pairs as input to a large pre-trained model is not novel (Wu et al., 2019; Zhang et al., 2019; Lee et al., 2019), as also pointed out by the authors, but the authors do not explicitly clarify this in the methodology section, leading the reader to believe that the domain-pairs is their own contribution. Same for the slot-gate (Wu et al., 2019).
>
> $\rightarrow$ Thank you for pointing out our unclear explanation. However, we think that our model is different from the models that you have mentioned in different ways. In all the models that you have mentioned (Wu et al., 2019; Zhang et al., 2019; Lee et al., 2019), relationship modeling among the different domain-slot pairs is not modeled. These models take input each individual domain-slot pair and predict the slot value for each domain-slot pair independently. Our model on the other hand, takes input all the domain-slot pairs which is represented as $\{DS\}$ tokens. As mentioned in section 2 related works, we believe that (Le et al., 2019) and (Chen et al., 2020)  are the two studies that directly models the relationship among different domain-slot pairs. However, our model differs from these models in that we used a pre-trained language encoder for predicting slot values.(Le et al., 2019) uses an independent Transformer based encoder. (Chen et al., 2020) uses a graph-attention network. We believe that by using the state-of-the-art pre-trained language encoder, along with the input representation we proposed using $\{DS\}$ tokens, enabled our model to achieve the state-of-the-art result in the MultiWOZ 2.1 dataset. For the slot-gate, we will mention that we did not invent the idea, but borrowed it from (Wu et al., 2019). Thank you for you suggestion to improve our article. We explicitly added this in section 3 Proposed method:
>
> > The adopted the concept of the slot-gate classifier from (Wu et al., 2019) and made adjustments to apply to our model.
>
>
>
> 2. The authors claim to learn relations between slots, but the analysis section is very thin and it just shows an ablation by masking the attention between the slot. Two points: why not just removing the [CLS] token instead of removing the attention, and why just using on ALBERTA large. For instance, the authors said "For this experiment, we used the ALBERT configuration of large-v2, for faster experimentation" which is contradictory since large-v2 is the slowest to run I guess. Can the authors show this ablation for all the model size?
>
> $\rightarrow$ Thank you for suggesting an ablation study to improve our manuscript. Would it be possible for you to make some clarifications on the ablation study? Does removing the $CLS$ token mean removing the $\{DS\}$ tokens and use the $CLS$ token only for the slot-value prediction? From our humble understanding, we think that removing the $\{DS\}$ tokens will make sure to prove that the model indeed need the $\{DS\}$ tokens to perform well. The experiment excluding $\{DS\}$ tokens is still running, but early results show that their performance is far inferior to the model witih $\{DS\}$ tokens (joint slot accuracy of 36% as of 6k steps of training). Please correct us if we misunderstood your ablation study suggestion.
>
> The reason why we experimented on ALBERT *large-v2* was due to lack of computational power. We were not able to run the model with *xlarge-v2*, which is the main model that is shown in Table 1.,  before the submission time. Since *large-v2* is smaller than *xlarge-v2*, we only ran the model with it. Currently, we are running the *xxlarge-v2* version for the ablation study of disabling attention among $\{DS\}$ tokens.
>
> Due to limitations to computing power, we expect that we would be unable to run ablation experiments for all configurations, but we will try our best.
>
> 3. Although, MWoZ is the current benchmark for DST in ToDs, there are also other datasets for this task that can be considered (e.g., Schema Guided Dialogue (SGD) (Rastogi et.al. 2019))
>
> $\rightarrow$ We agree that additional analysis on more datasets will prove the generalizability of our model. We will try to add experiments to those dataset if possible. However, since the SGD dataset is bigger than MultiWOZ 2.1, we unfortunately expect that it would be very hard to train the model on the SGD dataset.

---

> ### Author Response · Authors · 2020-11-13
> **Author Response to AnonReviewer2**
>
> Continued from previous comment.
>
> 4. [Reason to Reject] The main contribution of this paper is very thin, adding the [CLS] token as input, and the main technical contribution is not well explored (missing an in-depth ablation).
>
> $\rightarrow$ We believe that our contribution is adding $\{DS\}$ tokens (which work as domain-slot specific $CLS$ tokens) for the input representation and also using a pre-trained language encoder to encode the information using those  $\{DS\}$ tokens. We think that this idea is novel since as far as our knowledge, there has not been models running multiple $CLS$ tokens to represent different domain-slot pairs. Previous models that use pre-trained language encoders used only a single $CLS$ token by taking each single domain-slot pair information as the input and running the model multiple times. Our model can predict all domain-slot pair values in a single run, while modeling the dependencies among them. This results in a state-of-the-art result, which is achieved by using the *xxlarge-v2* version of ALBERT.
>
> However, we agree that our qualitative analysis is not comprehensive. We are running the ablation experiment that you have suggested, which we believe will prove that our proposed $\{DS\}$ does work very well. We will provide additional examples such as Fig 4.. Also, we will provide attention heat maps to infer how our model works.
>
> 5. [Reason to Accept] State-of-the-art performance at the submission time. To be noted, (Mehri et.al. 2020) reported better performance in MWoZ and other datasets, but this paper was released after the ICLR submission deadline.
>
> $\rightarrow$ Is the title of the (Mehri et.al. 2020) paper :Unsupervised Evaluation of Interactive Dialog with DialoGPT? From google scholar this paper was written by Mehri et al., and is related to dialog state tracking. However, the paper did not run experiments on the MultiWOZ 2.1 dataset. If I have misunderstood the paper you are mentioning please let us know.
>
> 5. Can the authors show the ablation for all the model size?
>
> $\rightarrow$ As mentioned in 2. of our comment, due to limitations to computing power, we expect that we would be unable to run ablation experiments for all configurations, but we will try our best.
>
> 6. Figure 4 is very hard to read. I suggest to better format the dialogue.
>
> $\rightarrow$ We agree that Fig 4. should be presented in a better way. We will add more examples and change the presentation of the analysis as well. Thank you very much for your suggestion to improve our manuscript.

---

### Official Review · AnonReviewer3 · 2020-10-27
**Interesting approach using pre-trained representations to encode domain-slot pairs for DST**

**Rating:** 7
**Confidence:** 4

**Review:**

Summary:
This paper showcases how pre-training can help with Dialogue State Tracking. The authors explicitly model
the relationship between domain-slot pairs. With their encoding and using strong pre-trained initializations
they are able to improve the joint goal accuracy by almost 1.5 points which is impressive.

Reasons for score:
This is a very well written paper and will be a good resource for people working on the task of Dialogue State Tracking.
The authors show how they can model relationships between domain-slot pairs and how they can encode them effectively
using pre-trained representations.
I am hoping that the authors can address some of the cons during the rebuttal period.

Pros:
1. Good dialogue representation which helps with the task of state tracking
2. Simple model consisting of encoders and 2 classifiers which are well explained.
3. Clear ablation study showing the value of 1) pre-training and 2) modeling relationship between domain-slot values


Cons:
1. This approach, like other popular approaches, suffers from the problem of having a fixed output vocabulary for slot values - hence limiting its scalability. While this cannot be addressed in this work, this is a drawback of this approach.
2. Some of the design decisions are stated but not well explained
- Only one pre-training method compared
- Authors mention they drop "dontcare" from slot gating but don't show the affect with or without it.
- Not much details on the setup and how it was trained.
3. Not much qualitative analysis.

Please address and clarify the cons above

Typos/Areas for improvement:
1. Section 3.2 and 3.3 can be shortened a lot. I would suggest showing more analysis.
- More examples of type of  mistakes fixed.
- Which turn in the dialogue does the error decrease the most.
- How much is the training time/ accuracy tradeoff
2. Adding another layer to make DS-split work should be trivial, there is no reason to leave that to future work.
Could you show how the results look with that?

------

Updating score based on authors' response.

---

> ### Author Response · Authors · 2020-11-13
> **Author Response to AnonReviewer3**
>
> Thank you very much for your review of our paper. We initially updated our paper with an updated result using *xxlarge-v2* of ALBERT, which showed the state-of-the-art result. We are running additional experiments to respond to the reviewers comment, as soon as the experiments are done we will update the manuscript right away.
>
> 1. This approach, like other popular approaches, suffers from the problem of having a fixed output vocabulary for slot values - hence limiting its scalability. While this cannot be addressed in this work, this is a drawback of this approach.
>
> $\rightarrow$ Thank you for pointing out the fundamental limitations of our approach. Yes we agree with this. This paper is using the fixed-vocabulary approach. However our **xxlarge-v2** version showed much better performance that any other model. We would like to think our model as the stepping stone to the generative models with open vocabulary. We currently working on how to deal with the open-vocabulary approach on the next paper.
>
> 2. Some of the design decisions are stated but not well explained: Only one pre-training method compared
>
> $\rightarrow$ For additional pre-trained encoders, we will try to run both BERT and Roberta. However due to the tight deadline, we think we would be possible to run only Roberta for the new pretrained encoder. We will update our manuscript as soon as the results come up.
>
> 3. Some of the design decisions are stated but not well explained: Authors mention they drop "dontcare" from slot gating but don't show the affect with or without it.
> $\rightarrow$ We will mention the precision/recall of the dontcare slot gating. In our preliminary models with ALBERT \textit{large-v2}, the prediction and recall for $dontcare$ was $48.87$% and $17.21$%, respectively. The precision and recall for $none$ showed $98.91$%, $99.45$% and $prediction$ $96.16$%, $94.93$%, respectively. We will mention this in the manuscript Section 3.2 Slot-gate classifier
>
> > In our preliminary models with ALBERT *large-v2*, the prediction and recall for dontcare was 48.87% and 17.21%, respectively. The precision and recall for none showed 98.91%, 99.45% and prediction 96.16%, 94.93%, respectively.
>
>
> 4. Some of the design decisions are stated but not well explained: Not much details on the setup and how it was trained.
>
> $\rightarrow$ We will describe in more detail on the experimental setup in section 4.2 Setup on the effective batch size and the GPU we used for training. We will provide more details to the setup if there are other details that you would like to know.
>
> 5. Not much qualitative analysis.
> $\rightarrow$ We agree that our analysis is not comprehensive. We are running additional experiments with only the $CLS$ token involved, without any $\{DS\}$ tokens, which was recommended by AnonReviewer2. This can show that involving  tokens is an effective solution. Early results show that the model without  tokens show very poor performance. As soon as the results are available, we will update our manuscript. We will also update our manuscript with additional examples of such improvements as well as some attention heat maps which can be used to infer how the model process.
>
> 6. Section 3.2 and 3.3 can be shortened a lot. I would suggest showing more analysis.
> $\rightarrow$ We will try to fit our paper in the 9 page limit with more analysis as mentioned above. We will try to shorten the Section that you have mentioned.
>
> 7. Which turn in the dialogue does the error decrease the most.
> $\rightarrow$ Could you please clarify what you mean by this comment? Does it mean to find a specific turn $t$ for some dialogue example and show qualitatively why that turn improved the error for that specific dialogue? We will run the analysis as you tell us to do.
>
> 8. How much is the training time/ accuracy tradeoff
> $\rightarrow$ We will provide the training time in terms of training steps and its corresponding accuracy. We did not record the results for early experiments with small models, so we are running them again. As soon as they are finished we will add the detail in the manuscript. For the *xxlarge-v2* version, in 20k steps 59.62%, in 25k steps: 63.46% 150k steps: $69.23$%.
>
> 9. Adding another layer to make DS-split work should be trivial, there is no reason to leave that to future work. Could you show how the results look with that?
> We were able to run the results with the smaller *base-v2* version. The results show that adding additional linear layers after concatenation did not improve the results. However, since this is a preliminary result, we will run the experiments on at least the *large-v2* version for confirmation. Note that in the *xxlarge-v2* version, the $DS_{split}$ worked better than the $DS_{merge}$ version. We think that this suggests that what matters is the encoding process of self-attention, not mere linear layers, which is in line with our preliminary results with *base-v2*.

---

### Official Review · AnonReviewer1 · 2020-10-29
**limited analysis**

**Rating:** 3
**Confidence:** 5

**Review:**

Summary:

This paper proposed a new approach for modeling multi-domain dialogue state tracking by incorporating domain-slot relationship using a pre-trained language encoder. The proposed approach are based on using special tokens to mode l such relationship. Two kinds of special tokens are proposed to represent domain-slot pair, DS_merge token for each specific pair, and tokens for every domain and slots separately



Comments:

1- what is the role of segment embedding (Fig3) for DST tank? Are two different segment used in the pretraining of model?

2- Table 1:
                  2-1 what type of label cleaning is used to compute joint goal accuracy? From SimpleTOD paper, each baseline has used different label cleaning
                  2-2: DS-split is bolded, while it is lower than SimpleTOD

3- Table 2: the results indicate that DST performance drops in total by blocking attention across different DS tokens. However, it is not clear how much of the performance drop belongs to turns with cross-domain related slots. The figure 4 only present one example of this case, which might not be correct for all wrong predictions. Also, it is helpful to report DST for single domain and to evaluate the importance of proposed approach.

4- Since the proposed approach can be used on any pretrained encoder, the evaluation on BERT and/or Roberta is helpful to understand the robustness of approach to the choice of pretrained encoder.



--------------------------------------------------------------
Post rebuttal:

Table 3:
the results indicates that only Albert-xxlarge achieve very high performance (222M). however, comparable models to other approaches, such as Roberta-base or albert-xlarge achieved around ~57% performance which is within margin of previous arts. for example, SimpleTOD with gpt2-base (124M) achieved 55.7% and ConvBert achieved 58%.
Therefor, it is unclear why Albert-xxlarge get so much higher performance compared to other encoders, since same tokenization and domain-slot relation is used.
Based on results in Table 3, there is inconsistency in which domain-slot relation does not always results in better performance,  and it depends on the choice of encoder too.

Overall, the proposed architecture is very similar to TRADE model, in terms of using an encoder for dialogue history, slot-gate and slot-value classifier. The only difference is in using a much powerful pretrained encoder.

---

> ### Author Response · Authors · 2020-11-13
> **Author Response to AnonReviewer1**
>
> Thank you very much for your review of our paper. We initially updated our paper with an updated result using *xx-large-v2* of ALBERT, which showed the state-of-the-art result. We are running additional experiments to respond to the reviewers comment, as soon as the experiments are done we will update the manuscript right away.
>
> Now, we will try to explain each point that you have mentioned.
>
> 1. What is the role of segment embedding (Fig3) for DST tank? Are two different segment used in the pretraining of model?
>
> $\rightarrow$ The role of the segment embedding is the default segment embedding for ALBERT as is described in the original paper (Lan, Zhenzhong, et al. 2019). During the pre-training of ALBERT, two segment embeddings are used, which is used to differentiate the two segments of sentences. Two segment embeddings are used because the order of two sentence segments can be continuous or random. In our model, we use a single segment embedding for the whole $\{DS\}$ tokens and $CLS$ and dialog context tokens. This is because as the $CLS$ token is subject to the first segment in the original ALBERT, $\{DS\}$ also serve as special $CLS$ tokens for each domain-slot pair.
>
> 2. Table 1: 2-1 what type of label cleaning is used to compute joint goal accuracy? From SimpleTOD paper, each baseline has used different label cleaning 2-2: DS-split is bolded, while it is lower than SimpleTOD
>
> $\rightarrow$ For label cleaning we used the method of TRADE, which is the standard that most models follow. It is described in section 4.1 Dataset as
> > We follow the pre-processing explained in (Wu et al., 2019).
>
> In Table 1, the SimpleTOD results are for the version using the label cleaning of TRADE. We will erase the bold of DS-split for the xlarge case. However, since our xx-large version achieved the new state-of-the-art result, we are changing Table 1. accordingly.
>
>
> 3. Table 2: the results indicate that DST performance drops in total by blocking attention across different DS tokens. However, it is not clear how much of the performance drop belongs to turns with cross-domain related slots. The figure 4 only present one example of this case, which might not be correct for all wrong predictions. Also, it is helpful to report DST for single domain and to evaluate the importance of proposed approach.
>
> $\rightarrow$ We understand that our analysis is not fully comprehensive. We assumed that the performance drop was due to the blocking of attention because the only different point of the model is that we disabled the attention across different DS tokens. Other model configurations were the same.  We are running additional experiments with only the $CLS$ token involved, without any $\{DS\}$ tokens, which was recommended by AnonReviewer2. This can show that involving $\{DS\}$ tokens is an effective solution. Early results show that the model without $\{DS\}$ tokens show very poor performance. As soon as the results are available, we will update our manuscript.
> We will also update our manuscript with additional examples of such improvements as well as some attention heat maps which can be used to infer how the model process.
> Does reporting DST for single domain mean that we should report the accuracy for each domain in the multi-domain setting? Or does it mean that we should train our model for a single domain, for example, the restaurant domain, and evaluate it? We will run the experiments as you tell us to do.
>
> 3.  In the domain split/merge method, does the domain/slot used as vocabulary tag or word embedding?
>
> $\rightarrow$ No, the \$\{DS\}$ tokens initialized randomly without a vocabulary tag or word embedding. We will look into methods to effectively incorporate word embeddings without harming computation costs.
>
> 4. Since the proposed approach can be used on any pretrained encoder, the evaluation on BERT and/or Roberta is helpful to understand the robustness of approach to the choice of pretrained encoder.
>
> $\rightarrow$ We will try to run both BERT and Roberta. However due to the tight deadline, we think we would be possible to run only Roberta for the new pretrained encoder. We will update our manuscript as soon as the results come up.

---

### Official Review · AnonReviewer4 · 2020-10-30
**This paper studies the problem of multi-domain dialogue state tracking which is challenging.  It proposes a bert based model for multi-domain dialogue that effectively models the relationship among domain-slot pairs.  The experimental results show the model achieve sota performance on the MultiWOZ-2.1 dataset.**

**Rating:** 5
**Confidence:** 3

**Review:**

Pros
•	This paper incorporates the multi-domain domain-slot pairs into the bert input so that the relations between sentences, domain-slot are modeled.
Cons
•	It’s better to experiment on more dataset to prove the method’s effectiveness.
Comments
•	This paper   https://arxiv.org/abs/2006.01554 seems get better performance on the same dataset.
•	In the domain split/merge method, does the domain/slot used as vocabulary tag or word embedding?
•	The slot-gate classifier’s output is fed into the slot-value classifier, how does this affect the performance?

---

> ### Author Response · Authors · 2020-11-13
> **Author Response to AnonReviewer4**
>
> Thank you very much for your review of our paper. We initially updated our paper with an updated result using *xx-large-v2* of ALBERT, which showed the state-of-the-art result. We are running additional experiments to respond to the reviewers comment, as soon as the experiments are done we will update the manuscript right away.
>
> Now, we will try to explain each point that you have mentioned.
>
> 1. It’s better to experiment on more dataset to prove the method’s effectiveness.
>
> $\rightarrow$ We agree that additional analysis on more datasets will prove the generalizability of our model. We know that there are other datasets such as Schema Guided Dialogue (SGD) (Rastogi et.al. 2019). We will try to add experiments to those dataset if possible.
>
> 2. This paper https://arxiv.org/abs/2006.01554 seems get better performance on the same dataset.
>
> $\rightarrow$ Thank you for recommeding the paper with better performance on the dataset. We were able to run the experiment with a bigger version of ALBERT *xxlarge-v2*, which enabled our model to achieve the state-of-the-art result as of now. We added the result of the paper you've mentioned in Table 1.
>
> 3.  In the domain split/merge method, does the domain/slot used as vocabulary tag or word embedding?
>
> $\rightarrow$ No, the \$\{DS\}$ tokens initialized randomly without a vocabulary tag or word embedding. We will look into methods to effectively incorporate word embeddings without harming computation costs.
>
> 4. The slot-gate classifier’s output is fed into the slot-value classifier, how does this affect the performance?
>
> $\rightarrow$The input of the slot-gate classifier are the  \$\{DS\}$ tokens as shown in equation (7) and (8). The input for the slot-value classifier are the  \$\{DS\}$ with the $CLS$ token as shown in equation (10) and (11). The input is partially shared, but the output of the slot-gate classifier is not used in the slot-value classifier.  If the domain-slot is predicted to $none$, the output of the slot-value classifier is changed into $none$ regardless of the prediction of the slot-value classifier. We will add this detail at the end of Section 3.2 slot-gate classifier.

---

### Author Response · Authors · 2020-11-19
**Revision adding more experimental results for ablation studies**

We added additional experimental results for ablation studies.

Specifically, we ran more experiments for proving that our domain-slot relationship modeling is effective. In addition to the blocking of attention among domain-slots, we ran a model without any $\{DS\}$ tokens at all, with only a $CLS$ token. We also ran the experiment for $ALBERT-xxlarge-v2$, in addition to the results from the $ALBERT-large-v2$ version.

We also ran another pretrained language model, RoBERTa. Specifically, RoBERTa-base and RoBERTa-large was added. Both models showed strong results.

We are currently running the model on MultiWOZ 2.2 (Zang et. al., 2020), which improved the MultiWOZ 2.1 dataset. We will update the manuscript as soon as the experiments are finished.

---

### Comment · ~Jinwon_An1 · 2021-02-01
**withdrawal of paper**

We withdraw the current version of our work. We found issues with the evaluation of the dataset with large models. Thus, the current version of our paper is invalid.

---

### Decision · Program_Chairs · 2021-01-07
**Final Decision**

**Decision:**

Reject

**Comment:**

The authors explore modeling the relationship between domain-slot pairs in multi-domain dialogue state tracking via use of special tokens in pre-trained contextualized word embeddings (i.e., one special token for each domain-slot pair or special tokens for the domain and the slot that are merged). Beyond this, the basic architecture is very similar to the TRADE architecture (and papers that build on this general slot-gate + slot-value classifier) for the fixed vocabulary setting. Experiments are conducted on the MultiWOZ 2.1/2.2 datasets, demonstrating impressive improvements over recent results.

== Pros ==
+ They demonstrate that domain-slot interdependencies can be modeled through special tokens for use with pre-trained embeddings.
+ The top-line empirical results are impressive.

== Cons ==
- Lack of a deep dive on the empirical analysis to show precisely why/where the proposed method is working better than existing work.
- The methodological advance is minimal beyond using better pre-trained embeddings.
- Only one dataset when others exist and this is largely an empirical paper.
- The writing is rushed and reads like a 'late-breaking' paper.

Evaluating along the specified dimensions:
* Quality: The quality of the work was the primary concern of the reviewers. Specifically, this reads like a 'late breaking' paper where the table of results is impressive, but there isn't significant examination of the empirical results showing why/when it works relative to competing methods. Focusing just on Tables 2 & 3, much of the improvement is ostensibly really due to the more powerful embeddings. Contextualizing this wrt {SimpleTOD, TRADE, DSTQA, Picklist}, this appears a minor methodological innovation centered around the input embeddings. The empirical results are impressive, but may very well be a result of the more powerful pre-trained embeddings -- additional empirical analysis and discussion might be able to convince the reader otherwise, but is lacking here.
* Clarity: This is a very simple idea, so it should be easily understood by most familiar with the research area. That being said, the paper seems very rushed in general.
* Originality: This applies ideas used in many NLP applications to the dialogue-state tracking problem. As previously stated, the architecture is similar to several existing DST formulations -- where the core idea is to model slot-value interdependencies through the contextualized embeddings using special token. While not a trivial idea, it also is something that many could/would have put together. Until it is abundantly clear that this isn't really a study of how to apply larger pre-trained embeddings to DST problems, it isn't clear that this is a significant dialogue systems advance beyond the strong performance.
* Significance: As stated, this isn't a significant methodological advance. However, the empirical results appear very impressive -- although the reviewers expressed some concerns regarding the evaluation. Since this is largely empirical, one of the reviewers pointed out that additional relevant datasets now exist, which would significantly strengthen the case.

In summary, the empirical results appear impressive, ostensibly setting the SoTA. However, there were several concerns regarding the novelty of the approach, if it is actually working better due to the reasons stated, sufficient analysis of the empirical results, amongst other things. Thus, despite the impressive results, the consensus evaluation was that this work is not ready for publication in its current form (even if the top-line results should be disseminated).